# A Comparison between Perceptions of Psychiatric Outpatients and Psychiatrists Regarding Benzodiazepine Use and Decision Making for Its Discontinuation: A Cross-Sectional Survey in Japan

**DOI:** 10.3390/ijerph20075373

**Published:** 2023-04-03

**Authors:** Yumi Aoki, Masahiro Takeshima, Takashi Tsuboi, Eiichi Katsumoto, Ken Udagawa, Ken Inada, Koichiro Watanabe, Kazuo Mishima, Yoshikazu Takaesu

**Affiliations:** 1Psychiatric and Mental Health Nursing, St. Luke’s International University, Tokyo 104-0044, Japan; 2Department of Neuropsychiatry, Kyorin University School of Medicine, Tokyo 181-8611, Japan; 3Department of Neuropsychiatry, Akita University Graduate School of Medicine, Akita 010-8543, Japan; 4Katsumoto Mental Clinic, Osaka 543-0056, Japan; 5Community Mental Health & Welfare Bonding Organization, Chiba 272-003, Japan; 6Department of Psychiatry, Kitasato University School of Medicine, Kanagawa 252–0374, Japan; 7Department of Neuropsychiatry, Graduate School of Medicine, University of the Ryukyus, Okinawa 903-0215, Japan

**Keywords:** benzodiazepine, cross-sectional survey, psychiatric outpatient, treatment decision making, shared decision making, Japan

## Abstract

Background: Although long-term use of benzodiazepines and z-drugs (BZDs) is not recommended, little is known about the stakeholders’ perceptions. This study aimed to assess and compare the perceptions of BZD use and decision making regarding its discontinuation between psychiatric outpatients and psychiatrists. Methods: A cross-sectional survey was conducted. Results: Of 104 outpatients, 92% were taking hypnotics and 96% were taking anxiolytics for ≥a year, while 49% were willing to taper hypnotic/anxiolytics within a year of starting. Most psychiatrists felt that “patient and psychiatrist make the decision together on an equal basis” compared to patients (*p* < 0.001), while more patients felt that “the decision is (was) made considering the psychiatrists’ opinion” compared to psychiatrists (*p* < 0.001). Of 543 psychiatrists, 79% reported “patients were not willing to discontinue hypnotic/anxiolytic” whereas a certain number of patients conveyed “psychiatrists did not explain in enough detail about hypnotic/anxiolytic discontinuation such as procedure (18.3%), timing (19.2%), and appropriate condition (14.4%)”. Conclusion: The results suggest that the majority of psychiatric outpatients were taking hypnotic/anxiolytics for a long time against their will. There might be a difference in perceptions toward hypnotic/anxiolytic use and decision making for its discontinuation between psychiatric outpatients and psychiatrists. Further research is necessary to fill this gap.

## 1. Introduction

Benzodiazepines and z-drugs (BZDs) act as positive allosteric modulators of the benzodiazepine binding sites of GABAA receptors, and have hypnotic-based sedation and anti-anxiety effects [1]. These drugs are frequently used as hypnotics or anxiolytics for those with insomnia, general anxiety disorder (GAD), and panic disorder. They are also commonly prescribed for anxiety symptoms, and not just for insomnia and anxiety spectrum disorders, which might lead to potentially inappropriate medications [2]. Moreover, BZDs are commonly prescribed for individuals with psychiatric disorders, such as schizophrenia and bipolar disorder, to manage psychiatric symptoms, including agitation and aggression [3].

The long-term use of BZDs has disadvantages such as dependence, decline in cognitive function, and motor impairment, leading to hip fractures associated with falls [4,5,6,7]. With regard to the treatment of insomnia, clinical guidelines recommend cognitive behavioral therapy for insomnia (CBT-I) as the first-line treatment, and medications including BZDs should only be considered if CBT-I is ineffective or unavailable [8,9]. It has also been suggested that BZDs should only be used for a short period of up to four weeks [9]. Despite these evidence-based recommendations, BZDs continue to be repeatedly prescribed worldwide. 

The prescription of BZDs is higher in Japan than in Western countries, according to a 2010 United Nations report [10]. The country’s universal healthcare coverage, which provides easy access to medical care and ensures a low financial burden, may make it difficult for healthcare providers and users to be mindful not to prescribe or use unnecessary BDZ [11]. Given this situation, the Japanese government implemented medical fee reductions for BDZs use three times between 2012 and 2018 to promote appropriate use (1 April 2012, 1 April 2014, and 1 April 2018). However, according to a survey using a large-scale health insurance claims database conducted in 2021 by Takeshima et al., the mean duration of hypnotic prescriptions was 2.9 months, and 9.3% of patients were prescribed hypnotics for 12 months [11]. Moreover, an observational study using a Japan Medical Data Center dataset containing all medical fee data of health insurance service subscribers from 2015 to 2019 showed that the prescription of high-potency hypnotics (>15 mg/day diazepam) and anxiolytics (>2 mg/day flunitrazepam) generally remained unchanged [12]. These results suggest that the Japanese policies did not significantly affect long-term and high-dose BZD use. Consequently, safe tapering or discontinuation of BZD is a crucial and urgent issue in clinical settings.

In recent years, traditional patient–clinician interaction that focused on the “expert” healthcare provider informing the patient on the best course of action for a specific treatment has been replaced with patient-centered approaches, such as shared treatment decision making [13,14]. In line with this, prescribing behavior becomes more equal in concordance based on a partnership between the patient and clinician, where the reflection of the patient’s opinion on the medication regimen is fully considered during the decision-making process [15]. Patient participation in self-healthcare decisions is an ethical imperative [16] and should be recognized as a right [17]. Several countries and federal initiatives have promoted shared decision making as a desirable medical approach [18,19,20,21]. Japanese clinical guidelines for schizophrenia, depression, and social anxiety disorder also recommend implementing shared decision making in practice [22,23,24].

Patients are willing to be aware of their illness and participate in the decision-making process [25]. This tendency has no exception in psychiatry [26]. Such a patient-centered approach has the potential to reduce the overuse of treatment options, including choice of medication, which does not benefit everyone [27]. 

However, patient involvement in treatment decision making in clinical settings is not fully understood. The extent to which patients are involved in prescribing behaviors related to BZD use remains unclear. Considering the long-term and high-dose use of BZDs, patient involvement may not be implemented in psychiatry in Japan.

This study aimed to assess the perceptions of BZD use and decision making regarding discontinuation of BZDs between psychiatric outpatients and psychiatrists. Considering the long-term and high-dose use of BZDs, we hypothesized that there might be discrepancies between the outpatients and psychiatrists regarding BZD use and discontinuation.

## 2. Materials and Methods

### 2.1. Study Design

We performed a cross-sectional, anonymous survey.

For psychiatric outpatients, we included members of the Mental Health and Welfare Bonding Organization (COMHBO), which is a nonprofit community organization comprising individuals with mental illnesses, such as schizophrenia and bipolar disorder, their families, and formal/informal caregivers. An invitation was sent to individuals with mental illnesses (*n* = 740) on 25 October 2021 via COMHBO email listservs with a link to the as a Google Forms survey. Those who met the following criteria were invited to participate in this survey: aged 20 years or above and currently on hypnotic/anxiolytic regimen or having taken hypnotic/anxiolytic drugs in the past. A reminder email was sent after two weeks, and the survey was closed on 26 November 2021.

We included psychiatrists belonging to the Japanese Association of Neuro-Psychiatric Clinics (*n* = 1690). An invitation was sent to each psychiatrist on 10 January 2022 via email listservs of the association, which included a link to the as Google Forms survey. We also dispatched each psychiatrist the survey by post to enable them to respond by whichever mode they preferred. The survey was closed on 26 January 2022.

This study was approved by the Institutional Review Board of St. Luke’s International University (2021-604).

### 2.2. Questionnaire and Statistical Analysis

The variables for the questionnaire were developed based on a literature review and discussions within the steering group of this study, including psychiatrists, a psychiatric nurse, and an outpatient with mental illness. The questionnaire consisted of the following components: (1) demographics; (2) current situation related to hypnotic/anxiolytic use, such as experience, duration, and the number of hypnotic/anxiolytic medications taken; (3) perspectives of hypnotic/anxiolytic use (desirable timing of reduction after symptom improvement and permissible situation of its continuation); (4) decision making regarding the continuing/tapering of hypnotic/anxiolytic use; and (5) difficulties when attempting to reduce hypnotic/anxiolytic use in the past. Regarding each variable, anxiolytics were benzodiazepines and hypnotics included both benzodiazepines and z-drugs (Appendix A). We then used descriptive statistics (mean and frequency) to investigate each item. The chi-square (χ^2^) test was used to compare outpatients’ responses to those of psychiatrists’ regarding the decisions on continuing/tapering hypnotic/anxiolytics. The results were considered statistically significant at *p* ≤ 0.05.

## 3. Results

### 3.1. Demographics

In total, 104 psychiatric outpatients and 543 psychiatrists completed the survey. Table 1 shows the respondents’ demographic details and other characteristics.

### 3.2. Outpatients’ Reported Current Situation of Hypnotic/Anxiolytic Use

Overall, 52.9% of outpatient respondents reported that they had taken both hypnotics/anxiolytics, while 30.8% had hypnotics only and 16.4% had anxiolytics only. Regarding the duration of taking hypnotics/anxiolytics, 92.0% of outpatient respondents reported that they had taken hypnotics for over a year, and 95.8% had anxiolytics for more than one year (Table 2).

### 3.3. Perspectives of Hypnotic/Anxiolytic Use

#### 3.3.1. Desirable Timing of Reduction after Symptom Improvement

The most frequent response of the outpatients was “no need to reduce hypnotic/anxiolytic if there are no side effects” (23.1%), followed by “immediately after symptom improvement” (19.2%), and “within 3 months after symptom improvement” (16.4%). On the other hand, 49.16% of outpatients desired to reduce hypnotic/anxiolytic usage within 12 months after symptom improvement, who constitute the participants that selected one of the following four variables: “immediately” (19.2%), “within 3 months” (16.4%), “within 6 months” (3.9%), and “within 12 months(9.6%).”

The most desirable reduction timing of hypnotic/anxiolytic use for the psychiatrists after patient’s symptom improvement was “within 3 months after symptom improvement” (38.1%), followed by “within 6 months after symptom improvement” (22.5 %), and “immediately after symptom improvement” (14.2%). Overall, 85.5% of psychiatrists desired to reduce hypnotic/anxiolytic use immediately within 12 months after symptom improvement as a total of four variables: “immediately” (14.2%), “within 3 months” (38.1%), “within 6 months” (22.5%), and “within 12 months” (8.3%).

In comparing the two groups, some variables included discrepancies: more psychiatrists thought that hypnotic/anxiolytic reduction should occur “within 3 months” (*p* < 0.001) and “within 6 months” (*p* < 0.001) compared to outpatients, while more outpatients thought that the reduction should occur “after 12 months passed” (*p* < 0.001) and “no need to reduce if there are no side effects” (*p* < 0.001) compared to psychiatrists (Table 3).

#### 3.3.2. Permissible Situation of Hypnotic/Anxiolytic Continuation

The most frequent answer by both the outpatients and psychiatrists was “when I (the patient) am (is) still suffering from symptoms of insomnia or anxiety” (71.2% of outpatients, 69.8% of psychiatrists), followed by “when my (the patient’s) social functioning is disrupted” (39.4% of outpatients, 55.1% of psychiatrists) and “when I (the patient) still have (has) physical or mental problem, which led to the start of the medication administration” (31.7% of outpatients, 47.0% of psychiatrists).

In comparing the two groups, there were discrepancies in three variables: psychiatrists were more careful about “social functioning disruption” (*p* < 0.001) and “physical and mental problems, which led them to start to take the medication” (*p* < 0.001), and “the number of medications taken” (*p* < 0.001) compared to the patients (Table 3).

#### 3.3.3. Decisions Making Regarding Continuing/Tapering Hypnotic/Anxiolytic Use

The ranking of the outpatients and psychiatrists was consistent, as follows: “the outpatient and psychiatrist make (made) the decision together, on an equal basis” (44.1% of outpatients, 79.6% of psychiatrists), followed by “the decision is (was) made, considering the psychiatrist’s opinion” (31.7% of outpatients, 16.6% of psychiatrists) and “the decision is (was) made, considering the patient’s opinion”(24.0% of outpatients, 3.5% of psychiatrists). In contrast, there was a significant difference between the proportion of outpatient respondents and psychiatrists for each variable. Thus, our hypothesis that there would be a discrepancy between outpatients and psychiatrists was accepted. Most psychiatrists thought that they made decisions on continuing/tapering hypnotic/anxiolytics using shared decision making on an equal basis, while the majority of outpatients believed that they made a decision considering the psychiatrist’s opinion. (Table 4).

### 3.4. Difficulties When Trying to Reduce Hypnotic/Anxiolytic in the Past

The most frequent difficulty of the outpatients was “I could not reduce the medication because withdrawal symptoms appeared when I tried” (24.0%), followed by “I could not reduce the medication because of worsening symptoms” (23.1%) and “I never had any particular problem regarding the medication reduction” (21.2%). Moreover, the outpatients experienced the following difficulties because of insufficient related information: “I did not know how to reduce the medication” (18.3%), “I did not know when I should reduce the medication” (19.2%), and “I did not know the situation or condition where the medication could be reduced” (14.4%). 

The most frequent answer of the psychiatrists was “I could not reduce the medication because the patient was unwilling to reduce it” (78.8%), followed by “I could not reduce the medication because of worsening symptoms” (61.3%) and “I could not reduce the medication because withdrawal symptoms appeared when I tried” (37.6%).

In comparing the two groups, most items had discrepancies: more outpatients felt that they did not know “how to reduce the medication” (*p* < 0.001), “when I should reduce the medication” (*p* < 0.001), and “the situation or condition where the medication can be reduced” (*p* < 0.001) due to insufficient information compared to the psychiatrists. On the other hand, more psychiatrists felt that they could not reduce the medication because of “worsening symptoms” (*p* < 0.001) and “withdrawal symptoms appeared when tried” (*p* < 0.001) compared to the patients (Table 5).

## 4. Discussion

This is the first study to assess and compare perceptions of BZD use and decision making regarding its discontinuation between psychiatric outpatients and psychiatrists.

In this study, almost half of the psychiatric outpatients (49.1%) and the majority of the psychiatrists (85.5%) desired to reduce hypnotic/anxiolytic use within 12 months of symptom improvement. In particular, psychiatrists were more willing to implement medication reduction if they thought that the reduction should occur “within 3 months” (*p* < 0.001) and “within 6 months” (*p* < 0.001) compared to outpatients. Thus, the stakeholders’ preferences were in line with the recommendations described in several clinical guidelines, which recommend that hypnotics/anxiolytics should not be prescribed for a longer time [8,9,10]. However, against their will, more than 90% of the outpatients remained on the hypnotic/anxiolytic for over 12 months. Most psychiatrists thought they could not reduce the medication because the patient was unwilling to. Moreover, a certain number of the outpatients felt that they had not been provided with sufficient information regarding hypnotic/anxiolytic usage reduction. This suggests that there should be more emphasis on sharing related information with the patients. Heather et al. reported that those with insomnia who received a letter warning about the harms of long-term use of BZD hypnotics showed larger reductions in BZD consumption than those who did not [22]. Accordingly, sharing not only the advantages, but also the disadvantages of hypnotic/anxiolytic behavior with individuals taking them might lead to successful medication reduction. In this context, shared treatment decision making is expected to be promoted in the clinical setting. 

This study found discrepancies between the perspectives of outpatients and psychiatrists. Thus, our hypothesis was accepted. First, more psychiatrists perceived permissible situations for hypnotic/anxiolytic continuation in relation to social functioning and physical or mental problems, which led to the start of the medication, compared to the patients. Second, more psychiatrists found worsening psychiatric and withdrawal symptoms challenging while attempting to reduce medication compared to the patients. These results suggest that patients may not focus on symptoms and function, whereas psychiatrists do. Thus, these discrepancies mean that psychiatrists might fail to understand the patient’s preferences and values, which should be clarified through conversation with the patients in a clinical setting. Third, more psychiatrists felt that they had already implemented shared decision making, while more patients felt that the decision had been made considering the psychiatrists’ opinions. This discrepancy is crucial because SDM may not be as widely implemented as psychiatrists believe. Several studies have reported similar situations. For instance, Matthias et al. observed appointments in psychiatric outpatient services for medication management and found that psychiatrists initiated most of the decisions [28]. Verwijmeren et al. also indicated that the degree of objective patient involvement in psychiatric consultations was low [29]. Accordingly, both psychiatrists and patients must understand these situations and discrepancies, which suggests that patient participation to clarify their preferences and values regarding BZD discontinuation should be encouraged in the clinical setting.

Based on the results of this study, we propose some solutions for the current situation in Japan, where long-term and high-dose BZD use remains. First, >90% of the patients in this study had used hypnotic/anxiolytic medication for over a year. Other countries have already set limits on the duration of BZD prescriptions at four weeks [30,31,32,33,34,35,36]. Therefore, it may be necessary to review our medical system, which has no duration limit and allows for long-term BZD use. Second, the results revealed that some psychiatrists still preferred long-term BZD use and believed there was no need to reduce if there were no side effects. Twenty-three percent of patients also felt no need to reduce it. For instance, the health system could introduce regular warning signs on medical records if BDZ use is longer than four weeks to promote awareness of the preferred short-term use of BDZs. Third, our results showed that most psychiatrists believed BZD use should be reduced early after symptom improvement. We must develop a system to put this positive perspective into practice. For example, communication support tools, such as patient decision aids [37], might be useful in discussing whether to continue or discontinue BZDs during consultation. Moreover, an inter-professional approach should also be considered. Other healthcare providers, such as pharmacists and nurses, could support patients in deliberating whether to continue or discontinue BDZ medication by providing related information on each treatment option neutrally [38,39]. Thus, it may be desirable to adopt such a policy of patient involvement during BZD discontinuation and reduce reimbursement as a national policy.

This study had several limitations. First, outpatients and psychiatrists were recruited from different organizations. The findings might have been different if the study had been conducted with a combination of the psychiatrist in charge and their patients. Second, outpatients in this study had mental health conditions, such as schizophrenia and bipolar disorder. The results may have differed with patients under primary care because mental illness is considered a risk factor for the long-term use of BZDs [40,41,42]. Third, the sample size of psychiatrists was approximately five times higher than that of participants. Fourth, psychiatrists were older than the outpatients, and male psychiatrists predominated. Differences in age and sex may lead to different perspectives and attitudes. Thus, these sociodemographic factors may have affected the study outcomes. Fifth, we did not examine patients’ characteristics, such as medical history and past and current medications. Sixth, regarding the item ‘I could not reduce the medication because the patient was unwilling to reduce it’ as difficulties while attempting to reduce hypnotic/anxiolytic in the past, we did not collect the patients’ data. Thus, we could not compare the perceptions of the two parties. Nonetheless, this study is valuable in the current situation, wherein the issue of long-term BZD use remains unresolved.

## 5. Conclusions

This study assessed and compared the perceptions on BZD use and decision making regarding its discontinuation between psychiatric outpatients and psychiatrists. The results suggest that the majority of psychiatric outpatients were taking hypnotics/anxiolytics for a long time against their will. There might be a difference in perception toward hypnotic/anxiolytic use and decision making for its discontinuation between psychiatric outpatients and psychiatrists. Further research is required to address this gap.

## Figures and Tables

**Table 1 ijerph-20-05373-t001:** Demographic details and other characteristics of the respondents.

Variables	Outpatients *n* = 104	Psychiatrists *n* = 543	*p* Value *
Sex, *n* (%)			
Female	57 (54.8)	107 (19.7)	<0.001
Male	46 (44.2)	433 (79.7)	<0.001
Other	1 (1.0)	3 (0.6)	0.626
Age (years), *n* (%)			
20–29	3 (2.9)	1 (0.2)	0.001
30–39	22 (21.2)	11 (2.0)	<0.001
40–49	42 (40.4)	70 (12.9)	<0.001
50–59	31 (29.8)	153(28.2)	0.736
60–69	5 (4.8)	195(35.9)	<0.001
70–79	1 (1.0)	96(17.7)	<0.001
≥80	0 (0)	15(2.8)	0.086
N.A.	-	2 (0.4)	-
Service used/affiliation, *n* (%)			
Clinic	54 (51.9)	536 (98.7)	<0.001
Psychiatric hospital	34 (32.7)	3 (0.6)	<0.001
General hospital	8 (7.7)	2 (0.4)	<0.001
University hospital	8 (7.7)	0 (0)	<0.001
N.A.	-	2 (0.4)	
Psychiatric diagnosis, *n* (%) **			
Schizophrenia	47 (45.2)	-	-
Bipolar disorder	25 (24.0)	-	-
Major depressive disorder	19 (18.3)	-	-
Anxiety disorder	14 (13.5)	-	-
Developmental disorder	13 (12.5)	-	-
Insomnia	12 (11.5)	-	-
Other	10 (9.6)	-	-
Unknown	3 (2.9)	-	-

* Based on χ^2^ test, ** Multiple answers, N.A.: No answer.

**Table 2 ijerph-20-05373-t002:** Response of outpatients pertaining to the current situation of hypnotic/anxiolytic use.

Variable	
Experience of taking hypnotic/anxiolytic, *n* (%), *n* = 104	
Hypnotic only	32 (30.8)
Anxiolytic only	17 (16.3)
Both hypnotic/anxiolytic	55 (52.9)
Duration of taking hypnotic, *n* (%), *n* = 87	
<1 month	1 (1.2)
1–3 months	4 (4.6)
3–6 months	2 (2.3)
6–12 months	0 (0)
≥12 months	80 (92.0)
Duration of taking anxiolytic, *n* (%), *n* = 72	
<1 month	2 (2.8)
1–3 months	0 (0)
3–6 months	0 (0)
6–12 months	1 (1.4)
≥12 months	69 (95.8)
Numbers taking hypnotic, *n* (%), *n* = 87	
1	44 (50.6)
2	28 (32.2)
3	6 (6.9)
≥4	9 (10.3)
Numbers taking anxiolytic, *n* (%), *n* = 72	
1	40 (55.6)
2	20 (27.8)
3	4 (5.6)
≥ 4	8 (11.1)

**Table 3 ijerph-20-05373-t003:** Perspectives related to hypnotic/anxiolytic use.

Variables	Outpatients *n* = 104	Psychiatrists *n* = 543	*p* Value *
Desirable timing of hypnotic/anxiolytic reduction after symptom improvement, *n* (%)			
Immediately	20 (19.2)	77 (14.2)	0.186
Within 3 months	17 (16.4)	207 (38.1)	<0.001
Within 6 months	4 (3.9)	122 (22.5)	<0.001
Within 12 months	10 (9.6)	45 (8.3)	0.656
After 12 months passed	11 (10.6)	13 (2.4)	<0.001
No need to reduce if there are no side effects	24 (23.1)	26 (4.8)	<0.001
Other	10 (9.6)	44 (8.1)	0.609
Unknown/N.A.	8 (7.7)	9 (1.6)	<0.001
Permissible situation of hypnotic/anxiolytic continuation **			
When I (the patient) am (is) still suffering from symptoms of insomnia or anxiety	74 (71.2)	379 (69.8)	0.782
When my (the patient’s) social functioning is disrupted	41 (39.4)	299 (55.1)	<0.001
When I (the patient) still have (has) physical or mental problem, which led to start to take the medication	33 (31.7)	255 (47.0)	<0.001
When I (the patient) want(s) to continue the medication	32 (30.8)	121 (22.3)	0.062
When I (the patient) am (is) not suffering from side effects of the medication	29 (27.9)	149 (27.4)	0.926
When my (the patient’s) quality of life is worsened	25 (24.0)	93 (17.1)	0.095
When the number of medications I (the patient) am (is) taking is low	8 (7.7)	157 (28.9)	<0.001
Other	5 (4.8)	11 (2.0)	0.094
Unknown/N.A.	2 (1.9)	3 (0.6)	0.144

* Based on χ^2^ test, ** Multiple answers, N.A.: No answer.

**Table 4 ijerph-20-05373-t004:** Decision making regarding continuing/tapering hypnotic/anxiolytic use.

Variables	Outpatients *n* = 104	Psychiatrists *n* = 541	*p* Value *
The decision is (was) made, considering the patient’s opinion	33 (31.7)	90 (16.6)	<0.001
The outpatient and psychiatrist make (made) the decision together, on an equal basis	46 (44.2)	432 (79.6)	<0.001
The decision is (was) made, considering the psychiatrist’s opinion	25 (24.0)	19 (3.5)	<0.001

* Based on χ^2^ test.

**Table 5 ijerph-20-05373-t005:** Difficulties cited while attempting to reduce hypnotic/anxiolytic in the past (multiple answers).

Variable	Outpatients *n* = 104	Psychiatrists *n* = 543	*p* Value *
I could not reduce the medication because the patient was unwilling to reduce it	-	428 (78.8)	-
I did not know how to reduce the medication due to insufficient related information (provided by psychiatrist)	19 (18.3)	8 (1.5)	<0.001
I did not know when I should reduce the medication due to insufficient related information (provided by psychiatrist)	20 (19.2)	14 (2.6)	<0.001
I did not know the situation or condition, where the medication can be reduced due to insufficient related information (provided by psychiatrist)	15 (14.4)	25 (4.6)	<0.001
I could not reduce the medication because of worsening symptoms	24 (23.1)	333 (61.3)	<0.001
I could not reduce the medication because withdrawal symptoms (e.g., tremors, palpitations, anxiety) appeared when I tried to reduce	25 (24.0)	204 (37.6)	0.008
I never had any particular problem regarding the medication reduction	22 (21.2)	34 (6.3)	<0.001
I never reduced the medication	15 (14.4)	1 (0.2)	<0.001
Other	11 (10.6)	12 (2.2)	<0.001
N.A.	-	34 (6.3)	-

* Based on χ^2^ test, N.A.: No answer.

## Data Availability

The datasets are available from the corresponding author on reasonable request.

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
