# Peer review of "A Comparison between Perceptions of Psychiatric Outpatients and Psychiatrists Regarding Benzodiazepine Use and Decision Making for Its Discontinuation: A Cross-Sectional Survey in Japan"

_ijerph, 2023, doi:10.3390/ijerph20075373_

Round 1

Reviewer 1 Report

Dear authors

Thank you for the opportunity to review the manuscript. This study examines various aspects of the use of benzodiazepines from the psychiatrists' and patients' perspectives, including the aspect of shared desicion making.

I think this topic is very important, benzodiazepines are used inflationary and especially prescribed too long. Absurdly, this creates its own health problems. This is a problem for the individual patient, as well as for the health care system.

I appreciate short and concise manuscripts, but in this one the content seems to me to be somewhat lacking.

Introduction:

-Many benzodiazpines are also prescribed for agitation, aggression, and other symptoms. This is especially true for people with schizophrenia or bipolar disorder. Consequently, it is unsurprising that most of their patient population also had this diagnosis. Please also refer to this in the introduction.

-In Central Europe, general practitioners in particular also prescribe benzodiazepines. This can, of course, cause difficulties. Since this is a Japanese study, it is essential to understand the situation in Japan in order to be able to classify the results. Is benzodiazepine abuse a big problem in Japan? Are there many addicts? What is the attitude of society? Is it a secret subpopulation? Do only psychiatrists prescribe benzodiazpeine or other doctors as well? Is there an active black market?

Methodology:

-Research results must be replicable. Please upload the questionnaire online and link to it, or make an appendix and include it.

-please formulate hypotheses when doing NHST tests (such as Chi squared). A test is only meaningful if there is a hypothesis and it is accepted/dismissed. Formulate your hypotheses and address whether it can be accepted or rejected in the results.

-I am generally missing the plot here from a statistical analytical point of view. Why do you compare absolute/relative numbers on most questions and then do a NHST on 3 selected questions (desicion making)? What is the rationale?

Discussion:

-As indicated in the introduction, I find the embedding of the results in their sociocultural context lacking. What do these results mean from a view of the japanese health system, how does it compare to other countries?

Regards

Author Response

Dear Reviewer 1,

Thank you for your careful review of our manuscript.

Your comments were highly insightful and enabled us to greatly improve the quality of our manuscript.

In accordance with all the suggestions, we have revised the manuscript using coloured text (red font).

We have attached point-by-point response describing how we have responded to all the reviewer’s suggestions.

Thank you again for your kind support.

Yours sincerely,

Authors

Reviewer 2 Report

Thank you for addressing this clinical issue in psychiatric and mental health settings. The study is organized and easy to flow, however there are few minor notes should be addressed before publications. 

1. Introduction: 

a) Need more explanation about the significance of the study? why you compare perceptions between psychiatric outpatients and psychiatrists regarding benzodiazepine use and decision- making for its discontinuation? 

b) There is no citations for references (9 and 10); that should be somewhere between lines 48 and 53. 

2. Materials and methods: 

a) it would be helpful if you are able to provide a brief description about the patients' characterstics.

3. Results:

a) Add (table 3) in line 146. 

b) line 150, the number should be consistent with table, therefore should be 44.2%.

c) lines 155-158 were interesting and need to be discussed deeply in your discussion. 

  d) lines 166-170 are confusing, it seems you add up three categories. this may confuse the readers. 

e) Table 5, the first raw (I could not reduce the medication because the patient was unwilling to reduce it), there is no data regarding patients, and you concentrate on that in your discussion. My point is: you cannot compare this statement between patients and psychiatrists. Therefore, you may reconsider this. 

4. Discussion 

a) the first appearance of (SDM in line 214), therefore spell out completely at the first time.   

b) in lines 197-199 (A majority of the psychiatrists (78.8%) answered that “I could not reduce the hypnotic/anxiolytic because patient was unwilling to reduce it” while almost half of the out-198 patients (49.1%) desired to reduce the hypnotic/anxiolytic usage within 12 months of 199 symptom improvement.). This is incomparable results. 

c) According to your conclusions and results, what do you recommend for patients and psychiatrists regarding clinical practice? 

Thank you  

Author Response

Dear Reviewer 2,

Thank you for your careful review of our manuscript.

Your comments were highly insightful and enabled us to greatly improve the quality of our manuscript.

In accordance with all the suggestions, we have revised the manuscript using coloured text (red font).

We have attached point-by-point response describing how we have responded to all the reviewer’s suggestions.

Thank you again for your kind support.

Yours sincerely,

Authors

Reviewer 3 Report

The topic of the study is very interesting and not an ordinary one. People nowadays do not think much about should the patient take part in treatment. Psychiatric treatment takes a very long time so this problem is actual. It's a good thing that the trend is toward having the important decisions for the patient made not only by the physician, but also by himself. It is interesting how the responses of psychiatrists and patients in this study differ. These differences suggest that the doctor-patient dialogue is not taking place at the appropriate level. However, there are shortcomings in the study . The most important is the mismatch between the two groups. Psychiatrists are significantly older than patients and male physicians are prevalent among them. This raises questions about all of the findings. Maybe the differences are related to the age of the participants rather than their physician/patient status. 

Minor:

Write the probabilities in Tables 1,2,3,5 as you have done in other Table 4.

Author Response

Dear Reviewer 3,

Thank you for your careful review of our manuscript.

Your comments were highly insightful and enabled us to greatly improve the quality of our manuscript.

In accordance with all the suggestions, we have revised the manuscript using coloured text (red font).

We have attached point-by-point response describing how we have responded to all the reviewer’s suggestions.

Thank you again for your kind support.

Yours sincerely,

Authors

Round 2

Reviewer 3 Report

Authors significantly improved their manuscript